# Association of the *IL-37* Polymorphisms with Transaminases and Alkaline Phosphatase Levels in Premature Coronary Artery Disease Patients and Healthy Controls. Results of the Genetics of Atherosclerotic (GEA) Mexican Study

**DOI:** 10.3390/diagnostics11061018

**Published:** 2021-06-02

**Authors:** Fabiola López-Bautista, Rosalinda Posadas-Sánchez, Gilberto Vargas-Alarcón

**Affiliations:** 1Department of Molecular Biology, Instituto Nacional de Cardiología Ignacio Chávez, Ciudad de México 14080, Mexico; nutrifabs@gmail.com; 2Department of Endocrinology, Instituto Nacional de Cardiología Ignacio Chávez, Ciudad de México 14080, Mexico; rossy_posadas_s@yahoo.it

**Keywords:** aminotransferases, inflammation, interleukin 37, polymorphisms, premature coronary artery disease

## Abstract

Interleukin 37 (*IL-37*) is an anti-inflammatory cytokine expressed in foam cells located in the atherosclerosis plaques. The present study aimed to evaluate the association of the *IL-37* polymorphisms with premature coronary artery disease (pCAD), cardiovascular risk factors, metabolic parameters, and levels of liver enzymes. Three *IL-37* polymorphisms (rs6717710, rs2708961, and rs2708947) were determined in 1161 patients with pCAD and 951 healthy controls. *IL-37* polymorphisms were not associated with the presence of pCAD. The association of the polymorphisms with cardiovascular risk factors, metabolic parameters, and levels of liver enzymes was evaluated independently in pCAD and healthy controls. In pCAD patients, under different models, the rs6717710 was associated with low risk of having elevated alkaline phosphatase (ALP) (*p*_additive_ = 0.020; *p*_dominant_ = 0.02; *p*_heterozygous_ = 0.04; *p*_codominant1_ = 0.040). On the other hand, in healthy controls, the rs6717710 was associated with low risk of having elevated levels of alanine aminotransferase (ALT) (*p*_additive_ = 0.04, *p*_recessive_ = 0.01, *p*_codominant2_ = 0.01) and aspartate aminotransferase (AST) (*p*_additive_ = 0.02, *p*_dominant_ = 0.02). The *IL-37* polymorphisms were not associated with the risk of pCAD. In pCAD patients, the rs6717710 was associated with low risk of having elevated ALP levels, whereas in controls was associated with low risk of having elevated ALT and AST levels.

## 1. Introduction

Coronary artery disease (CAD) is the principal consequence of the presence of atherosclerosis for a long time [1]. It is a progressive and chronic disease modulated by environmental and genetic factors [2]. Several studies have proposed liver enzymes as cardiovascular disease (CVD) markers [3,4,5]. Transaminases are not traditional cardiovascular risk factors; however, it has been observed that elevated concentrations of liver enzymes have been associated with mortality from cardiovascular disease. In addition, several recent cardiovascular studies have shown that liver enzymes gamma-glutamyl transferase (GGT), alkaline phosphatase (ALP), and aspartate aminotransferase (AST)/alanine aminotransferase (ALT) can be a good predictor of CVD mortality [6,7,8]. A meta-analysis of prospective cohort studies reported that GGT and ALP are positively associated linearly with CVD risk. Stratified analysis by cause-specific cardiovascular endpoints showed that ALT was inversely associated with coronary heart disease and positively with stroke. When the analysis was made in the different ethnic groups, a positive association of ALT with CVD in Asian populations was observed [9]. The atherosclerosis process begins when LDL particles penetrate the subendothelial space, where they are oxidized by reactive oxygen species [10]. These oxidized particles are later phagocytosed by resident macrophages that later form the so-called foam cells [10]. During this stage of the process, a significant number of inflammatory molecules, both pro- and anti-inflammatory, are produced at the injury site [11]. One of these molecules is the interleukin 37 (IL-37), which has been found expressed in foam cells within atherosclerotic plaques [12]. IL-37 is an anti-inflammatory cytokine belonging to the interleukin 1 family [12] with meaningful participation in the atherosclerotic process. It has been reported that IL-37 decreases the extracellular and intracellular inflammation acting through the IL-1 receptor family member 8 (IL-1R8) and the transcription factor Smad3, respectively [13]. Similarly, IL-37 has an essential effect on macrophages, reducing the production of various cytokines such as IL-1B, IL-6, and IL-12, cytokines with a critical role in atherosclerosis [14]. Thus, IL-37 functions as an essential inhibitor of innate immunity and inflammation [15]. High levels of IL-37 in patients with atrial fibrillation [16], arterial calcification [17], and acute coronary syndrome [18] have been reported. In animal models, the use of IL-37 has been beneficial for the treatment of myocardial infarction (MI) [19], ischemia/reperfusion injury [20], and vascular calcification [21]. In the same way, in an animal model, the treatment with IL-37 reduced proinflammatory cytokine production by hepatocytes and Kupffer cells, protecting against hepatic ischemia/reperfusion (I/R) injury. In this model, serum levels of ALT were markedly decreased with a reduction of approximately 34% [22]. The IL-37 is part of the *IL-1* gene cluster located on chromosome 2q12–13 and spans a 360-kb region [23]. The *IL-37* gene is polymorphic, and some of these polymorphisms have been associated with susceptibility to some diseases such as rheumatoid arthritis [24], autoimmune thyroid disease [25], Behcet’s disease [26], and systemic lupus erythematosus [27]. Regarding cardiovascular disease, Yin et al. reported the association of the *IL-37* rs3811047 polymorphism with increased CAD risk in two independent cohorts from China [28]. Considering the critical role of the IL-37 in the inflammatory process and its role in CAD and hepatic injury, the present study aimed to evaluate the association of the *IL-37* polymorphisms with premature coronary artery disease (pCAD), cardiovascular risk factors, metabolic parameters, and levels of liver enzymes in a cohort of patients well-characterized from the demographic, anthropometric, clinical, and biochemical point of view. Based on the bioinformatics analysis and previous results, we selected the rs6717710, rs2708961, and rs2708947 polymorphisms for the present study.

## 2. Material and methods

### 2.1. Study Population

This study is a cross-sectional analysis of the baseline stage of the GEA study previously described [29,30]. This analysis included 1161 patients with premature CAD and 951 control subjects (Figure 1).

PCAD was defined by the personal history of myocardial infarction, angioplasty, revascularization surgery, or coronary stenosis ≥ 50% detected by angiography; diagnoses were made before 55 in men and 65 in women. Patients were recruited in the Department of Hemodynamics at Instituto Nacional de Cardiología Ignacio Chávez in Mexico City. We excluded patients with acute cardiovascular events three months before the date of recruitment or congestive heart failure, because they are in an acute event with biochemical parameters altered. The control group included subjects with no evidence of coronary artery calcification and no family history of pCAD; these subjects were recruited from the blood bank donors and through written media invitation at social service centers. The total sample did not include individuals with a history or evidence of kidney, liver, thyroid, or oncology disease and those with corticosteroid treatment.

Standardized questionnaires were applied to obtain demographic information, schooling level, economic income, family and personal history of cardiovascular disease, dietary habits, physical activity, alcohol and smoking consumption, as well as drug use. BMI was calculated through the formula (weight, kg)/(size, m^2^). Blood pressure was determined after a period of rest, with the digital sphygmomanometer WelchAllyn, series 52,000 (Skaneateles Falls, NY, USA); the average of the last two of three consecutive measurements was used for analysis. Diagnosis of hypertension was defined as ≥ 140/90 mmHg or the use of antihypertensive treatment [31]. 

Hypercholesterolemia was defined as total cholesterol (TC) > 200 mg/dL; high LDL-c were considered when LDL-c > 130 mg/dL, hypoalphalipoproteinemia if HDL-c < 40 mg/dL in men and < 50 mg/dL in women [32], and hypertriglyceridemia if Tg ≥ 150 mg/dL [33]. American Diabetes Association criteria were used to define diabetes mellitus [34]. 

The cut-off point for defining elevated liver enzymes was considered when the value was found above the 75th percentile for ALT ≥ 23 IU/L in women and ≥30 IU/L in men, AST ≥ 27 IU/L in women and ≥ 29 IU/L in men, and ALP ≥ 90.2 in women and ≥ 83 in men. These cut-off points were obtained from a subsample of the GEA study that included 131 men and 185 women without obesity and normal blood pressure, glucose, and lipids.

Coronary artery calcium (CAC) was quantified by computed tomography and CAC scoring according to the Agatston method [35] for defined the control group (CAC = 0 Agatston Units). Abdominal fat was evaluated with the use of the Kvist method [36] and the liver-to-spleen attenuation rate (L:SAR) according to Longo et al. [37] and after the hepatic steatosis was defined as L:SAR ≥ 1.0 [38]. Only individuals with CAC equal to zero were included in the control group.

The study was approved by the Institutional Ethics Committee and following the guidelines of the Helsinki Declaration. All participants signed the informed consent.

### 2.2. Bioinformatics Analysis

The possible functional effect of the *IL-37* polymorphisms was analyzed using the following bioinformatics tools: FastSNP [39], SNP Function Prediction [40], Human-transcriptome Database [41], Splice Port: An Interactive Splice Site Analysis Tool [42], SNPs3D [43], ESE finder [44], and HSF [45].

### 2.3. Genetic Analysis

Genomic DNA from peripheral blood containing EDTA was isolated using QIAamp DNA Blood Mini kit (QIAGEN, Hilden, Germany). On the same day the blood was taken, the DNA was extracted. Polymorphisms rs6717710, rs2708961, and rs2708947 were genotyped according to the manufacturer’s instructions (Applied Biosystems, Foster City, CA, USA), using 5′ exonuclease TaqMan genotyping assays on an ABI Prism 7900HT Fast Real-Time PCR system.

### 2.4. Statistical Analysis

We use median with interquartile range (25th to 75th percentile), mean ± standard deviation, and percentage to present the data. According to the data distribution, the comparison between the groups was evaluated with the Student’s *t*-test or Mann–Whitney U test for continuous and chi-square test for categorical variables. All polymorphisms were in Hardy–Weinberg equilibrium (*p* > 0.05). Logistic regression models were constructed for each polymorphism and inheritance model separately, additive (major allele homozygotes vs. heterozygotes vs. minor allele homozygotes), dominant (major allele homozygotes vs. heterozygotes + minor allele homozygotes), recessive (major allele homozygotes + heterozygotes vs. minor allele homozygotes), heterozygous (heterozygotes vs. major allele homozygotes + minor allele homozygotes), co-dominant1 (heterozygotes vs. major allele homozygotes), and codominant2 (major allele homozygotes vs. minor allele homozygotes), to assess whether any polymorphism is associated independently with pCAD, cardiovascular risk factors, metabolic parameters, or levels of liver enzymes. The models were adjusted with biological relevance variables and those that showed a statistically significant difference between the groups. *p* < 0.05 was considered statistically significant. Statistical package STATA/MP: Release 16 (College Station, TX, USA: Stata Corp LLC) was used for the analysis. 

## 3. Results

### 3.1. Clinical and Biochemical Characteristics

We included 1161 patients with pCAD and 951 control subjects. In the total population, 54% were male (41% in controls and 81% in patients), with an average age of 52.8 ± 8.5 years (controls 51.5 ± 8.9; cases: 54.1 ± 8.1) and BMI 28.5 ± 4.2 kg/m^2^, for both groups. Characteristics of pCAD patients and healthy controls are shown in Table 1. In patients with pCAD, age, visceral abdominal fat, triglycerides, glucose, and insulin concentrations were significantly higher (*p* < 0.001) compared to controls, as well as the higher frequency of use of lipid-lowering (97.4% in patients and 14.6% in controls), hypoglycemic (34.5% in patients and 7.78% in controls), and antihypertensive therapy (97.7% in patients and 10.3% in controls). The lipid-lowering drugs used by the patients were statins (91.3%), fibrates (15%), cholesterol absorption inhibitors (3.7%), and a low proportion used niacin (0.5%). Regarding hypoglycemic agents, patients used oral hypoglycemic (30.1%) and insulin therapy (7.8%). The antihypertensive drugs used were beta-blockers (85.4%), ACE inhibitors (75.2%), angiotensin-II receptor antagonists (13.3%), diuretics (19.3%), and calcium channel blocker (16.1%) as monotherapy or combination therapies.

In contrast, the concentrations of total cholesterol, LDL-c, HDL-c, apoB, apoA, hs-CRP, liver enzymes, and smoking were significantly lower in pCAD patients (*p* < 0.001). The low levels of TC in patients can be due to the statin treatment that they receive. 

Cardiovascular risk factors’ prevalence in pCAD patients and controls is shown in Figure 2. As expected, the prevalence of obesity, hypertension, hypoalphalipoproteinemia, hypertriglyceridemia, diabetes, and insulin resistance were significantly higher in the pCAD patients (*p* < 0.001). In contrast, the control group showed a higher frequency of hypercholesterolemia and elevated LDL-c (*p* < 0.001).

### 3.2. Association of IL-37 Polymorphisms with pCAD

Allele and genotype distribution of the three study polymorphisms in pCAD patients and healthy controls and their possible functional effect are shown in Table 2. PCAD patients and healthy controls showed a similar distribution of allele and genotype.

According to the logistic regression analysis, no association with pCAD was detected (Figure 3).

### 3.3. Association of the IL-37 Polymorphisms with Cardiovascular Risk Factors, Metabolic Parameters, and Levels of Liver Enzymes

The association of the polymorphisms with cardiovascular risk factors, metabolic parameters, and levels of liver enzymes was evaluated independently in pCAD patients and healthy controls (Figure 4). In healthy controls, the rs6717710 polymorphism was associated with low risk of presenting elevated ALT (*p*_additive_ = 0.04; *p*_recessive_ = 0.01; *p*_codominant2_ = 0.01) and elevated AST (*p*_additive_ = 0.02; *p*_dominant_ = 0.02). A moderate association of the rs2708947 polymorphism with a low risk of hypercholesterolemia was observed (*p*_heterozygous_ = 0.05; *p*_codominant1_ = 0.05). On the other hand, in pCAD patients, the rs6717710 was associated with low risk of presenting elevated ALP (*p*_additive_ = 0.02; *p*_dominant_ = 0.02; *p*_heterozygous_ = 0.04; *p*_codominant1_ = 0.04). Models were adjusted by age, sex, BMI, diabetes, LDL-c, hypertension, and smoke.

## 4. Discussion

IL-37 is an anti-inflammatory cytokine with meaningful participation in the atherosclerotic process. The *IL-37* gene is polymorphic, and some polymorphisms have been associated with the development of inflammatory and autoimmune diseases. Considering that inflammation has an essential role in the atherosclerotic process, we analyzed the distribution of three *IL-37* polymorphisms in patients with pCAD. The polymorphisms were not associated with pCAD. However, the rs6717710 was associated with low risk of presenting elevated ALP levels in pCAD patients and with low risk of presenting elevated ALT and AST levels in healthy controls.

Recently, Yin et al., analyzed the distribution of the *IL-37* rs3811047 polymorphism in patients with CAD of two independent cohorts from Northern and Central China [28]. The authors reported that the rs3811047 *A* allele was significantly associated with CAD in both cohorts and the combined populations. Our study did not include the rs3811047 polymorphism because, in the informatics analysis, this polymorphism was not functional. Instead of this polymorphism, we analyzed the rs6717710, which according to our informatics analysis is possibly functional, creating a binding site for the GATA6 transcriptional factor, and is in complete linkage disequilibrium with the rs3811047 polymorphism. In our cohort of patients, the rs6717710 polymorphism was not associated with pCAD. Inclusion criteria for patients and controls could explain the difference between the study of Yin and ours. In Yin’s study, patients with CAD were included, while we included patients with pCAD (diagnosis was made before age 65 in women and 55 in men). On the other hand, we included in the study healthy controls without familial antecedents of coronary disease. This information is not reported in the study by Yin et al. [28]. Differences in the distribution of polymorphisms may also explain the differences between the studies. Data reported from the National Center for Biotechnology Information showed that individuals with Mexican (Los Angeles) and Asian Ancestry together with Mexican mestizo individuals (13.5%) included in the present study have a lower frequency of the rs6717710 *C* allele (16% and 20%, respectively) compared with Caucasian and Africans (30% and 85%, respectively). In the same way, the Mexican mestizos (3.4%) of our study, together with Caucasian, Africans, and individuals from Los Angeles with Mexican ancestry, have a lower frequency of the rs2708961 *C* allele (8%, 18%, and 6%, respectively), whereas in the Asian population this allele is null (0%). Concerning the rs2708947 polymorphism, Mexican mestizos of our study together with individuals from Los Angeles with Mexican Ancestry, Caucasians, and Africans present the lower frequencies of the *C* allele (3.4%, 6%, 8%, and 17%, respectively) [46,47].

In our study, an association of the *IL-37* polymorphisms with ALP (rs6717710 in pCAD patients), ALT (rs6717710 in controls), and AST (rs6717710 in controls) levels was observed. In a previous study by our group, an association of the *IL-37* rs2708961 and rs2708947 polymorphisms with a low risk of hypercholesterolemia was reported [48]. In the present study, a moderate association of the rs2708947 polymorphism with hypercholesterolemia was observed, corroborating the association of this polymorphism reported by Lopez-Bautista et al., [48]. ALT and AST catalyze the transfer of amino groups to generated products in gluconeogenesis and amino acid metabolism, and they are produced abundantly by hepatocytes [49,50]. On the other hand, ALP catalyzes the hydrolysis of inorganic pyrophosphate, which is a vascular calcification inhibitor [5]. Some prospective epidemiological studies have demonstrated an association between ALP, ALT, and AST levels and cardiovascular diseases [5,51,52]. Tonelli et al. suggest that the association of ALP with cardiovascular diseases could be due to the promotion of vascular calcification and its pro-inflammatory effects [53]. Genome-wide association and candidate gene studies have defined some polymorphisms associated with AST, ALT, and ALP [54,55,56,57,58,59]. One of the genes most repeatedly associated with the transaminases (ALT and AST) is the one that codes for the patatin-like phospholipase 3, known as adiponutrin [57,58,59]. Adiponutrin is expressed in white adipose tissue and the liver [60] and has been related to obesity [61]. Another gene associated with ALT is the helix–loop–helix ubiquitous kinase (*CHUK*), a gene related to glucose and lipid metabolisms [54,55]. Finally, the *ABO* locus has been associated with ALP [54,62,63]. Thus, our study is the first that reported an association of *IL-37* polymorphisms with low risk of having high levels of ALT, AST, and ALP. The mechanism by which IL-37 could affect liver enzyme levels is unknown; however, the anti-inflammatory effect of this cytokine could be regulating the levels of these hepatic enzymes. Li et al., analyzing the IL-37 concentrations in patients with chronic hepatitis B and C virus, establish that individuals with high ALT and AST levels also presented higher levels of IL-37 [64]. The authors suggest that the IL-37 as an anti-inflammatory cytokine could reduce excessive inflammatory damage in these patients. The determination of these enzymes is simple, inexpensive, and with assays sensitive and well standardized. Its evaluation and its association with genetic polymorphisms are essential, because they are emerging risk markers for CVD and could be significant in its prediction and prevention.

The study’s strengths include analyzing a well-characterized cohort from a clinical, demographic, biochemical, and anthropometric perspective. Knowledge of these variables allowed us to adjust the association analysis for possible confounding variables. It is crucial to consider that the control group used in the study only included individuals with CAC equal to zero, eliminating those individuals with subclinical atherosclerosis (CAC > 0) that could bias the results. Some limitations should be considered; we only analyzed three polymorphisms of the *IL-37* gene; however, according to our bioinformatics analysis, these polymorphisms had a possible functional effect. The rs6717710 and rs2708961 are located in the promoter region and create binding sites for GATA, GATA6, BRCA, and MYB transcription factors; rs2708947 polymorphism located in the exon 3 produces an amino acid change (arginine by tryptophan). IL-37 levels were not determined, and its association with the studied polymorphisms was not evaluated. The study was not replicated in an independent cohort of individuals. Finally, considering the genetic characteristics of the Mexican population, the results would be applied only in this population; thus, studies in other populations with different genetic backgrounds are mandatory.

## 5. Conclusions

In summary, the results suggest that the *IL-37* polymorphisms studied are not associated with pCAD in the Mexican population. However, the rs6717710 polymorphism is associated with a low risk of presenting high ALP levels in pCAD patients and high ALT and AST levels in healthy controls. 

## Figures and Tables

**Figure 1 diagnostics-11-01018-f001:**
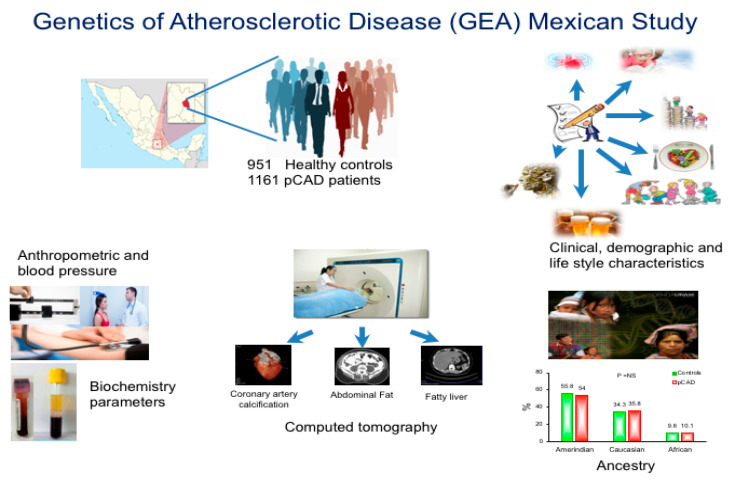
Study design and methodology.

**Figure 2 diagnostics-11-01018-f002:**
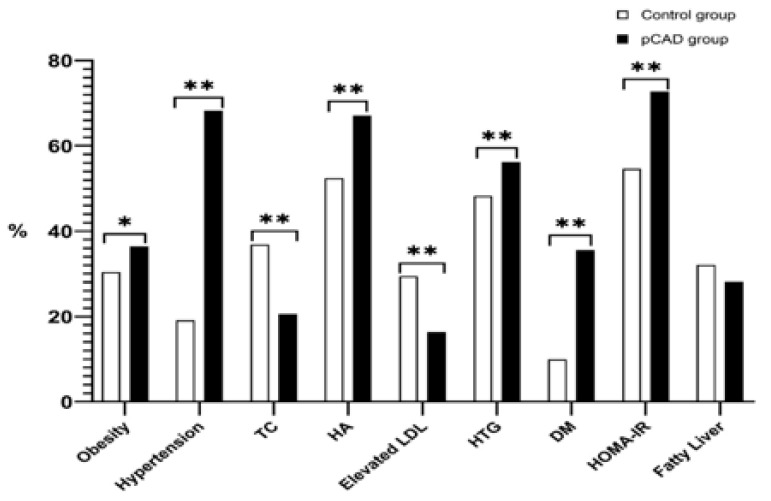
Prevalence of cardiovascular risk factors in the population studied. Data are presented in percentages. Significant *p* value * <0.05; ** <0.001. TC = total cholesterol; HA = Hypoalphalipoproteinemia; Elevated LDL ≥ 130 mg/dL; DM = diabetes; HOMA-IR = Homeostatic Model Assessment for Insulin Resistance.

**Figure 3 diagnostics-11-01018-f003:**
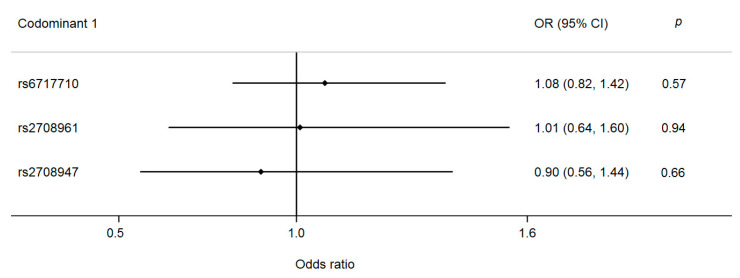
Association of the *IL-37* gene polymorphisms with pCAD. Models were adjusted by age, sex, BMI, diabetes, LDL-c, hypertension, and smoke. Codominant 1 model = heterozygous vs. major allele homozygous.

**Figure 4 diagnostics-11-01018-f004:**
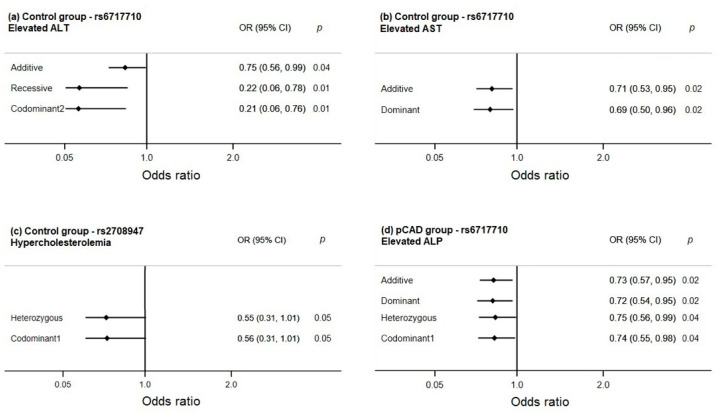
Association of the *IL-37* polymorphisms with transaminases and alkaline phosphatase. (**a**) The rs6717710 was associated with low risk of elevated ALT in controls, (**b**) the rs6717710 was associated with low risk of elevated AST in controls, (**c**) the rs2708947 was associated with low risk of hypercholesterolemia in controls, (**d**) the rs6717710 was associated with low risk of elevated ALP in pCAD patients. Models were adjusted by age, sex, BMI, diabetes, LDL-c, hypertension, and smoke. AST: Aspartate aminotransferase; ALT: Alanine aminotransferase; ALP: Alkaline phosphatase.

**Table 1 diagnostics-11-01018-t001:** Clinical and biochemical characteristics of pCAD patients and healthy controls.

Characteristic	Controls	pCAD	*p*
	n = 951	n = 1161	
Sex (% male)	41	81	<0.001
Age (years)	51.5 ± 8.9	54.1 ± 8.1	<0.001
Body mass index (kg/m^2^)	28.2 ± 4.1	28.8 ± 4.3	0.56
Systolic blood pressure (mmHg)	114 ± 16	118.7 ± 18.6	0.0003
Diastolic blood pressure (mmHg)	71 ± 8.7	72.1 ± 9.7	0.006
Visceral adipose fat (cm^2^)	141 (05–180)	170 (130–218)	<0.001
Subcutaneous adipose fat (cm^2^)	285 (215–368)	248 (194–318)	<0.001
Total adipose fat (cm^2^)	432 (345–535)	428 (340–532)	0.01
Total cholesterol (mg/dL)	191.1 ± 35.8	166.4 ± 47.7	<0.001
Triglycerides (mg/dL)	146.2 (108–204)	161.5 (118.2–219.4)	<0.001
HDL-c (mg/dL)	46.4 ± 13.7	38.8 ± 10.5	<0.001
LDL-c (mg/dL)	116.6 ± 31.1	96.1 ± 39.1	<0.001
Glucose (mg/dL)	96.4 ± 30.3	111.8 ± 43.4	<0.001
Insulin (μU/mL)	16.9 (12.3–22.7)	18.9 (13.8–26.5)	<0.001
HOMA-IR	3.7 (2.58–5.48)	4.8 (3.3–7.2)	<0.001
apoB (mg/dl)	93.5 (76–113)	79 (63–101)	<0.001
apoA (mg/dl)	137.2 ± 36.1	121.4 ± 26.6	<0.001
ALT (IU/L)	24 (18–34)	26 (19–36)	0.017
AST (IU/L)	25 (21–30)	26 (22–32)	0.001
ALP (IU/L)	80 (67–96)	77 (64–95)	0.001
hs-CRP (mg/L)	1.49 (0.8–3.0)	1.19 (0.64–2.73)	0.0001
Smoke			
Current n (%)	208 (21.8)	111 (11.6)	<0.0001
Past n (%)	323 (33.9)	622 (65.4)	
Never n (%)	420 (44.1)	218 (22.9)	
Tobacco index	0.15 (0–1.65)	0.65 (0.05–1.65)	<0.0001
Lipid lowering n (%)	139 (14.6)	1131 (97.4)	<0.001
Hypoglycemic n (%)	74 (7.78)	401 (34.5)	<0.001
Antihypertensive therapy n (%)	98 (10.3)	1135 (97.7)	<0.001

Data are presented as mean ± SD, median (interquartile range). Significant value of *p* < 0.05: t-student, U-Mann-Whitney, and Chi2. HOMA-IR: homeostasis model assessment–insulin resistance; apoB: apolipoprotein B; apoA: apolipoprotein A; hs-CRP: High sensitivity C reactive protein; ALT: Alanine aminotransferase; AST: Aspartate aminotransferase; ALP: Alkaline phosphatase; HDL-c: High density lipoprotein cholesterol; LDL-c: Low density lipoprotein cholesterol.

**Table 2 diagnostics-11-01018-t002:** Characteristics of *IL-37* polymorphisms in the present study.

rs Number Genotypes	Chr	Chr Position	MAF * (Control/Case)	Genotype Frequencies (%) *		Localization	Effect **
				Controls	Cases		
rs6717710 (*TT/TC/CC*)	2	112912187	0.135/0.133	74.9/23/2.0	74.8/23/1.6	Promotor	*C* → GATA or GATA6
rs2708961 (*TT/TC/CC*)	2	112912248	0.034/0.036	93.1/6.7/0.1	93/6.7/0.2	Promotor	*T* → BRCA, MYB
rs2708947 (*TT/TC/CC*)	2	112918642	0.034/0.033	93.3/6.4/0.2	93.3/6.5/0.09	Exon 3	*T* → Arginine *C* → Tryptophan

* Data of the present study. ** Results of bioinformatics analyses. Transcription factors: GATA, GATA6, BRCA, and MYB.

## Data Availability

The data shown in this article are available upon request from the corresponding author.

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
