# Peer review of "Association of the *IL-37* Polymorphisms with Transaminases and Alkaline Phosphatase Levels in Premature Coronary Artery Disease Patients and Healthy Controls. Results of the Genetics of Atherosclerotic (GEA) Mexican Study"

_diagnostics, 2021, doi:10.3390/diagnostics11061018_

Round 1
Reviewer 1 Report
The authors analyzed association of the IL-37 polymorphisms with transaminases and alkaline phosphatase levels in premature coronary artery disease and in healthy controls. Presented data seems to have low value and not appropriate for the Journal. Furhtermore, small study group from single center might be important limitation.
- What is clinical importance of this study? Any impact on everyday clinical practise?
- Was data evaluated on other populations ?
Author Response
1.- What is clinical importance of this study? Any impact on everyday clinical practice?
Answer: In order to clarify these comments of the reviewer, the following phrases have been added
in the introduction section:
“Several studies have proposed liver enzymes as CVD markers [3-5]. A meta-analysis of prospective cohort studies reported that GGT and ALP are positively associated linearly with CVD risk. Stratified analysis by cause-specific cardiovascular endpoints showed that ALT was inversely associated with CHD and positively with stroke. When the analysis was made in the different ethnic groups, a positive association of ALT with CVD in Asian populations was observed [6]”
“In the same way, in an animal model, the treatment with IL-37 reduced proinflammatory cytokine production by hepatocytes and Kupffer cells, protecting against hepatic ischemia/reperfusion (I/R) injury. In this model, serum levels of ALT were markedly decreased with a reduction of approximately 34% [19].”
In the introduction section, the phrase “Considering the critical role of the IL-37 in the inflammatory process, the present study aimed to evaluate the association of the IL37 polymorphisms with premature coronary artery disease (pCAD) and cardiometabolic parameters in a cohort of patients well-characterized from the demographic, anthropometric, clinic, and biochemical point of view.” has been modified to “Considering the critical role of the IL-37 in the inflammatory process, its role in CAD, and in hepatic injury, the present study aimed to evaluate the association of the IL37 polymorphisms with premature coronary artery disease (pCAD), cardiovascular risk factors, and levels of liver enzymes in a cohort of patients well-characterized from the demographic, anthropometric, clinic, and biochemical point of view.”
The following references have been added.
Monami, M.; Bardini, G.; Lamanna, C.; Pala, L.; Cresci, B.; Francesconi, P.; Buiatti, E.; Rotella, C.M.; Mannucci, E. Liver enzymes and risk of diabetes and cardiovascular disease: results of the Firenze Bagno a Ripoli (FIBAR) study. Metabolism 2008;57:387e92.
Yun, K.E.; Shin, C.Y.; Yoon, Y.S.; Park, H.S. Elevated alanine aminotransferase levels predict mortality from cardiovascular disease and diabetes in Koreans. Atherosclerosis 2009;205:533e7.
Wannamethee, S.G.; Sattar, N.; Papcosta, O.; Lennon, L.; Whincup, P.H. Alkaline phosphatase, serum phosphate, and incident cardiovascular disease and total mortality in older men. Arterioscler. Thromb. Vasc. Biol. 2013;33:1070e6.
Kunutsor, S.K.; Apekey, T.A.; Khan, H. Liver enzymes and risk of cardiovascular disease in the general population: a meta-analysis of prospective cohort studies. Atherosclerosis. 2014;236:7–17.
Sakai, N.; Van Sweringen, H.L.; Belizaire, R.M.; Quillin, R.C.; Schuster, R.; Blanchard, J.; Burns, J.M.; Tevar, A.D.; Edwards, M.J.; Alex B Lentsch, A.B. Interleukin-37 reduces liver inflammatory injury via effects on hepatocytes and non-parenchymal cells. J. Gastroenterol. Hepatol. 2012;27:1609–1616.
The phrase “The determination of these enzymes is simple, inexpensive, and with assays sensitive and well standardized. Its evaluation and its association with genetic polymorphisms are essential because they are emerging risk markers for CVD and could be significant in its prediction and prevention.” has been added in the discussion section.
Reviewer 2 Report
The authors in this study wanted to evaluate the association of IL-37 polymorphism with premature coronary artery disease and cardiovascular risk factors in a population of patients and in a group of healthy subjects.
Comments
1) Line 56: coronaru in coronary
2) Caption of the figure Fig 1: Description of the population studied and of the tests performed
3) In the methods the authors should explain well why they chose transaminases as cardiovascular risk factors, also presenting them as indices of liver damage.
4) Paragraph 2.2: in the text it would not be appropriate to include internet Link. Eventually they could be included in the references
5) Line 135: Why did you choose student's t test or Mann-Whitney U test? Based on what do you choose one or the other? Shouldn't the same test always be used?
6) Why does the control group show significantly higher cholesterol levels? Did the pCAD patient group take statins? Pharmacological therapy is not mentioned in the study.
7) Paragraph 3.3 should be rewritten better. It is difficult to read. I would recommend describing the results in a table.
8) Fig 3 shows the results of various polymorphisms in association with pCAD. The authors had previously only talked about three polymorphisms. So where do these results come from?
9) Figure 4 shows the significant associations between polymorphisms and high enzymatic levels or hypercholesterolemia without ever referring to a dominant, recessive, additive, etc. Authors should better explain their meaning based on a given polymorphism. Without further clarification, the figure is not easy to understand.
10) Line 241-242: the authors should explain well what they mean: ……….not functional and perhaps benign. However I think it is not correct to study the rs6717710 polymorphism claiming to arrive at the same conclusions as with the rs3811047 polymorphism. So the comparison with the results of the study by Yin et al. is difficult to interpret.
11) Line 265: generally it does not seem correct to me insert a link in the text of the manuscript, as already mentioned above. Better to insert it in the references
12) Line 269 “These results establish that the polymorphisms studied in our population are not directly associated with pCAD but with risk factors for this disease.” Instead in paragraph 3.3 it is reported that the studied polymorphisms were associated with a low risk of presenting high levels of transaminase or hypercholesterolemia. It seems like an inconsistency. Explain better.
13) Line 305-307: unclear. Explain well what functional effect means. It is also not clear what the authors mean by: “the possible association of these polymorphisms with IL-37 levels was not analyzed because these levels were not determined in the study.”
Author Response
1) Line 56: coronaru in coronary
Answer: It has been corrected
2) Caption of the figure Fig 1: Description of the population studied and of the tests performed
Answer: The caption of figure 1 has been modified as suggested by the reviewer.
Fig 1: Study design and methodology
3) In the methods the authors should explain well why they chose transaminases as cardiovascular risk factors, also presenting them as indices of liver damage.
Answer: Although transaminases have been associated with increased mortality from cardiovascular diseases, they are not considered a cardiovascular risk factor, due to this in the text we refer to them as liver enzymes.
4) Paragraph 2.2: in the text it would not be appropriate to include internet Link. Eventually they could be included in the references
Answer: The internet links have been added in the reference list.
5) Line 135: Why did you choose student's t test or Mann-Whitney U test? Based on what do you choose one or the other? Shouldn't the same test always be used?
Answer: We used both tests, the student's t test for normal distributed data and the Mann-Whitney U test for no normal distributed data.
In order to clarified this point, the phrase “The comparison between the groups was evaluated with the Student's t-test or Mann-Whitney U test for continuous and chi-square test for categorical variables.” has been modified to “According to the data distribution, the comparison between the groups was evaluated with the Student's t-test or Mann-Whitney U test for continuous and chi-square test for categorical variables.”
6) Why does the control group show significantly higher cholesterol levels? Did the pCAD patient group take statins? Pharmacological therapy is not mentioned in the study.
Answer: As the reviewer comments, once diagnosed CAD, the patients are treated with statins to lower cholesterol levels. This is the reason for why the patients had lower levels of total cholesterol. The phrase “In contrast, the concentrations of total cholesterol, LDL-c, HDL-c, apoB, apoA, hs-CRP, liver enzymes, and smoking were significantly lower (p<0.001).” has been changed to “In contrast, the concentrations of total cholesterol, LDL-c, HDL-c, apoB, apoA, hs-CRP, liver enzymes, and smoking were significantly lower in pCAD patients (p<0.001). The low levels of total cholesterol in patients can be due to the statin treatment that they receive.” in results section.
7) Paragraph 3.3 should be rewritten better. It is difficult to read. I would recommend describing the results in a table.
Answer: In order to clarified this point, the paragraph only include the p value for each significant comparison.
Paragraph: “The association of the polymorphisms with cardiovascular risk factors and metabolic parameters was evaluated independently in pCAD patients and healthy controls (Figure 4). In healthy controls the rs6717710 polymorphism was associated with low risk to present elevated ALT (padditive = 0.04; precessive = 0.01; pcodominant2 = 0.01), and elevated AST (padditive = 0.02; pdominant = 0.02). A moderate association of the rs2708947 polymorphism with a low risk of hypercholesterolemia was observed (pheterozygous = 0.05; pcodominant1 = 0.05). On the other hand, in pCAD patients, the rs6717710 was associated with low risk to present elevated ALP (padditive = 0.02; pdominant = 0.02; pheterozygous = 0.04; pcodominant1 = 0.04). Models were adjusted by age, sex, BMI, diabetes, LDL-C, hypertension, and smoke.”
The footnote of figure 4 was modified indicating the group and the associated polymorphism.
Figure 4. Association of the IL-37 polymorphisms with transaminases and alkaline phosphatase. (a). The rs6717710 was associated with low risk of elevated ALT in controls, (b). The rs6717710 was associated with low risk of elevated AST in controls, (c). The rs2708947 was associated with low risk of hypercholesterolemia in controls, (d). The rs6717710 was associated with low risk of elevated ALP in pCAD patients. Models were adjusted by age, sex, BMI, diabetes, LDL-C, hypertension, and smoke. AST: Aspartate aminotransferase; ALT: Alanine aminotransferase; ALP: Alkaline phosphatase.
8) Fig 3 shows the results of various polymorphisms in association with pCAD. The authors had previously only talked about three polymorphisms. So where do these results come from?
Answer: This was a mistake, the fig 3 has been modified and now only shown the 3 polymorphisms analyzed.
9) Figure 4 shows the significant associations between polymorphisms and high enzymatic levels or hypercholesterolemia without ever referring to a dominant, recessive, additive, etc. Authors should better explain their meaning based on a given polymorphism. Without further clarification, the figure is not easy to understand.
Answer: In order to clarify this point, the models have been described in the statistical analysis section. The phrase “Logistic regression models were constructed for each polymorphism and inheritance model separately: additive (major allele homozygotes vs. heterozygotes vs. minor allele homozygotes), dominant (major allele homozygotes vs. heterozygotes + minor allele homozygotes), recessive (major allele homozygotes + heterozygotes vs. minor allele homozygotes), heterozygous (heterozygotes vs. major allele homozygotes + minor allele homozygotes), co-dominant1 (heterozygotes vs. major allele homozygotes), and codominant2 (major allele homozygotes vs. minor allele homozygotes), to assess whether any polymorphism is associated independently with pCAD, cardiovascular risk factors or metabolic parameters.” has been added.
10) Line 241-242: the authors should explain well what they mean: ……….not functional and perhaps benign. However I think it is not correct to study the rs6717710 polymorphism claiming to arrive at the same conclusions as with the rs3811047 polymorphism. So the comparison with the results of the study by Yin et al. is difficult to interpret.
Answer: Considering that the term benign is confused, the phrase has been changed to “Our study did not include the rs3811047 polymorphism because, in the informatics analysis, this polymorphism was not functional.”
In order to clarify the phrase related with the study by Yin et al., the phrase “Considering that both polymorphisms are in strong linkage disequilibrium, similar associations with CAD could have been detected in both studies.” has been deleted.
11) Line 265: generally it does not seem correct to me insert a link in the text of the manuscript, as already mentioned above. Better to insert it in the references
Answer: The internet links have been added in the reference list.
12) Line 269 “These results establish that the polymorphisms studied in our population are not directly associated with pCAD but with risk factors for this disease.” Instead in paragraph 3.3 it is reported that the studied polymorphisms were associated with a low risk of presenting high levels of transaminase or hypercholesterolemia. It seems like an inconsistency. Explain better.
Answer: We agree with the reviewer and we considered that this phrase is not necessary and it has been deleted.
13) Line 305-307: unclear. Explain well what functional effect means. It is also not clear what the authors mean by: “the possible association of these polymorphisms with IL-37 levels was not analyzed because these levels were not determined in the study.”
Answer: The functional effect has been described and the phrase “Some limitations should be considered; we only analyzed three polymorphisms of the IL-37 gene; however, according to our bioinformatics analysis, these polymorphisms had a possible functional effect.” has been changed to “Some limitations should be considered; we only analyzed three polymorphisms of the IL-37 gene; however, according to our bioinformatics analysis, these polymorphisms had a possible functional effect. The rs6717710 and rs2708961 are located in the promoter region and create binding sites for GATA, GATA6, BRCA, and MYB transcription factors, rs2708947 polymorphism located in the exon 3 produces an amino acid change (arginine by tryptophan).”
On the other hand, the phrase “the possible association of these polymorphisms with IL-37 levels was not analyzed because these levels were not determined in the study.” was modified to “IL-37 levels were not determined, and its association with the studied polymorphisms were not evaluated.”
Reviewer 3 Report
Minor comments
Please provide information on timeframe between blood collection and DNA isolation.
Please provide justification for the exclusion of patients, e.g. line 78
Please provide a bit more background - why this study is performed, what is the novelty of this study? This should be included into the introduction.
Author Response
1.- Please provide information on timeframe between blood collection and DNA isolation.
Answer: The timeframe between blood collection and DNA isolation has been added. The phrase “On the same day, the blood was taken, the DNA was extracted.” has been added in material and methods section.
2.- Please provide justification for the exclusion of patients, e.g. line 78
Answer: These individuals were excluded because they are in acute event in which all the biochemical parameters are altered. The phrase “We excluded patients with acute cardiovascular events three months before the date of recruitment or congestive heart failure.” has been modified to “We excluded patients with acute cardiovascular events three months before the date of recruitment or congestive heart failure because they are in an acute event with biochemical parameters altered.”
3.- Please provide a bit more background - why this study is performed, what is the novelty of this study? This should be included into the introduction.
Answer: In order to clarify these comments of the reviewer, the following phrases have been added:
“Several studies have proposed liver enzymes as CVD markers [3-5]. A meta-analysis of prospective cohort studies reported that GGT and ALP are each positively associated in a linear fashion with CVD risk. Stratified analysis by cause-specific cardiovascular endpoints showed that ALT was inversely associated with CHD and positively with stroke. When the analysis was made in the different ethnic groups, a positive association of ALT with CVD in Asian populations was observed [6]”
“In the same way, in an animal model, the treatment with IL-37 reduced proinflammatory cytokine and chemokine production by hepatocytes and Kupffer cells, protecting against hepatic ischemia/reperfusion (I/R) injury. In this model, serum levels of ALT were markedly decreased with a reduction by approximately 34% [19].”
In the introduction section, the phrase “Considering the critical role of the IL-37 in the inflammatory process, the present study aimed to evaluate the association of the IL37 polymorphisms with premature coronary artery disease (pCAD) and cardiometabolic parameters in a cohort of patients well-characterized from the demographic, anthropometric, clinic, and biochemical point of view.” has been modified to “Considering the critical role of the IL-37 in the inflammatory process, its role in CAD, and in hepatic injury, the present study aimed to evaluate the association of the IL37 polymorphisms with premature coronary artery disease (pCAD), with cardiovascular risk factors, and with levels of hepatic enzymes in a cohort of patients well-characterized from the demographic, anthropometric, clinic, and biochemical point of view.”
Monami, M.; Bardini, G.; Lamanna, C.; Pala, L.; Cresci, B.; Francesconi, P.; Buiatti, E.; Rotella, C.M.; Mannucci, E. Liver enzymes and risk of diabetes and cardiovascular disease: results of the Firenze Bagno a Ripoli (FIBAR) study. Metabolism 2008;57:387e92.
Yun, K.E.; Shin, C.Y.; Yoon, Y.S.; Park, H.S. Elevated alanine aminotransferase levels predict mortality from cardiovascular disease and diabetes in Koreans. Atherosclerosis 2009;205:533e7.
Wannamethee, S.G.; Sattar, N.; Papcosta, O.; Lennon, L.; Whincup, P.H. Alkaline phosphatase, serum phosphate, and incident cardiovascular disease and total mortality in older men. Arterioscler. Thromb. Vasc. Biol. 2013;33:1070e6.
Kunutsor, S.K.; Apekey, T.A.; Khan, H. Liver enzymes and risk of cardiovascular disease in the general population: a meta-analysis of prospective cohort studies. Atherosclerosi. 2014;236:7–17.
Sakai, N.; Van Sweringen, H.L.; Belizaire, R.M.; Quillin, R.C.; Schuster, R.; Blanchard, J.; Burns, J.M.; Tevar, A.D.; Edwards, M.J.; Alex B Lentsch, A.B. Interleukin-37 reduces liver inflammatory injury via effects on hepatocytes and non-parenchymal cells. J. Gastroenterol. Hepatol. 2012;27:1609–1616.
The phrase “The determination of these enzymes is simple, inexpensive and with assays sensitive and well standardized. Its evaluation and its association with genetic polymorphisms are important because they are emerging risk markers for CVD, and could be important in its prediction and/or prevention.” has been added in the discussion section.
Round 2
Reviewer 1 Report
no further comments. Revision was prepared in satisfactory manner.
Author Response
No corrections were a request
Reviewer 2 Report
Authors should clarify some of the answers they gave:
Number 3:
In the methods the authors should explain well why they chose transaminases as cardiovascular risk factors, also presenting them as indices of liver damage.
Answer: Although transaminases have been associated with increased mortality from cardiovascular diseases, they are not considered a cardiovascular risk factor, due to this in the text we refer to them as liver enzymes.
And then the authors should explain why they used them even if they do not believe they are cardiovascular risk factors. If the authors consider liver function-related transaminase levels they should better explain why they evaluated them in association with IL-37 polymorphisms in pCAD patients.
Number 5:
Line 135: Why did you choose student's t test or Mann-Whitney U test? Based on what do you choose one or the other? Shouldn't the same test always be used?
Answer: We used both tests, the student's t test for normal distributed data and the Mann-Whitney U test for no normal distributed data.
Why didn't the authors choose to transform data that did not follow a normal distribution and use the student's T-test?
Number 6:
Why does the control group show significantly higher cholesterol levels? Did the pCAD patient group take statins? Pharmacological therapy is not mentioned in the study.
Answer: As the reviewer comments, once diagnosed CAD, the patients are treated with statins to lower cholesterol levels. This is the reason for why the patients had lower levels of total cholesterol. The phrase “In contrast, the concentrations of total cholesterol, LDL-c, HDL-c, apoB, apoA, hs-CRP, liver enzymes, and smoking were significantly lower (p<0.001).” has been changed to “In contrast, the concentrations of total cholesterol, LDL-c, HDL-c, apoB, apoA, hs-CRP, liver enzymes, and smoking were significantly lower in pCAD patients (p<0.001). The low levels of total cholesterol in patients can be due to the statin treatment that they receive.” in results section.
However, the authors should describe in detail in the text the pharmacological therapies of pCAD patients.
Author Response
Reviewer 2
Authors should clarify some of the answers they gave:
Number 3:
And then the authors should explain why they used them even if they do not believe they are cardiovascular risk factors. If the authors consider liver function-related transaminase levels they should better explain why they evaluated them in association with IL-37 polymorphisms in pCAD patients.
Answer:
We wish to clarify that transaminases are not traditional cardiovascular risk factors; however, it has been observed that elevated concentrations of liver enzymes have been associated with mortality from cardiovascular disease. In addition, several recent cardiovascular studies have shown that liver enzymes (GGT, ALP, and AST / ALT) can be a good predictor of CVD mortality. These articles have been detailed in the introduction. On the other hand, the relationship of the IL-37 with liver injury and with ALT levels was previously reported in an animal model with IL-37 treatment,
The phrase “Transaminases are not traditional cardiovascular risk factors; however, it has been observed that elevated concentrations of liver enzymes have been associated with mortality from cardiovascular disease. In addition, several recent cardiovascular studies have shown that liver enzymes gamma-glutamyl transferase (GGT), alkaline phosphatase (ALP), aspartate aminotransferase (AST)/alanine aminotransferase (ALT) can be a good predictor of CVD mortality [6-8]. A meta-analysis of prospective cohort studies reported that GGT and ALP are positively associated linearly with CVD risk. Stratified analysis by cause-specific cardiovascular endpoints showed that ALT was inversely associated with coronary heart disease and positively with stroke. When the analysis was made in the different ethnic groups, a positive association of ALT with CVD in Asian populations was observed [9].” was included in the introduction section.
The following references have been included:
6.- Rahmani, J.; Miri, A.; Namjoo, I.; Zamaninour, N.; Maljaei, M.B.; Zhou, K.; Cerneviciute, R.; Mousavi, S.M.; Varkaneh, H.K.; Salehisahlabadi, A.; Zhang, Y. Elevated liver enzymes and cardiovascular mortality: A systematic review and dose-response meta-analysis of more than one million participants. Eur. J. Gastroenterol. Hepatol. 2019, 31, 555–562.
7.- Choi, K.M.; Han, K.; Park, S.; Chung, H.S.; Kim, N.H.; Yoo, H.J.; Seo, J.A.; Kim, S.G.; Kim, N.H.; Baik, S.H.; Park, Y.G.; Kim, S.M. Implication of liver enzymes on incident cardiovascular diseases and mortality: A nationwide population-based cohort study. Sci. Rep.2018, 8, 3764.
8.- Porter, S.A.; Pedley, A.; Massaro, J.M.; Vasan, R.S.; Hoffmann, U.; Fox, C.S. Aminotransferase levels are associated with cardiometabolic risk above and beyond visceral fat and insulin resistance: The framingham heart study. Arterioscler. Thromb. Vasc. Biol. 2013, 33, 139–146.
The phrase “In the same way, in an animal model, the treatment with IL-37 reduced proinflammatory cytokine production by hepatocytes and Kupffer cells, protecting against hepatic ischemia/reperfusion (I/R) injury. In this model, serum levels of ALT were markedly decreased with a reduction of approximately 34% [19].” was added in the introduction section.
Number 5:
Why didn't the authors choose to transform data that did not follow a normal distribution and use the student's T-test?
Answer:
In order to make the characteristics of the study population easier to interpret, we decided to use non-parametric tests. It would have been helpful to carry out the logarithmic transformation of the variables if we had performed linear regression analysis; however, this was not the case in the present study.
Number 6:
Why does the control group show significantly higher cholesterol levels? Did the pCAD patient group take statins? Pharmacological therapy is not mentioned in the study.
However, the authors should describe in detail in the text the pharmacological therapies of pCAD patients.
Answer: The detailed description of the drugs used by the patients has been included. The phrase “In patients with pCAD, age, visceral abdominal fat, triglycerides, glucose, and insulin concentrations were significantly higher (p < 0.001) compared to controls; as well as the higher frequency of use of lipid-lowering (97.4% in patients and 14.6% in controls), hypoglycemic (34.5% in patients and 7.78% in controls) and antihypertensive therapy (97.7% in patients and 10.3% in controls). The lipid-lowering drugs used by the patients were statins (91.3%), fibrates (15%), cholesterol absorption inhibitors (3.7%), and a low proportion used niacin (0.5%). Regarding hypoglycemic agents, patients used oral hypoglycemic (30.1%) and insulin therapy (7.8%). The antihypertensive drugs used were beta-blockers (85.4%), ACE inhibitors (75.2%), angiotensin-II receptor antagonists (13.3%), diuretics (19.3%), and calcium channel blocker (16.1%); as monotherapy or combination therapies.” has been added in the results section.
In table 1 has been added a line with data of antihypertensive therapy in patients and controls.